# Cardiac Behavior and Heart Rate Variability in Elderly Hypertensive Individuals during Aerobic Exercise: A Non-Randomized Controlled Study

**DOI:** 10.3390/ijerph20021292

**Published:** 2023-01-11

**Authors:** Paulo Evaristo de Andrade, Juliana Zangirolami-Raimundo, Tassiane Cristina Morais, Luiz Carlos De Abreu, Carlos Eduardo Siqueira, Isabel Cristina Esposito Sorpreso, José Maria Soares Júnior, Rodrigo Daminello Raimundo

**Affiliations:** 1Laboratório de Delineamento de Estudos e Escrita Científica, Centro Universitário FMABC, São Paulo 09060-590, Brazil; 2Disciplina de Ginecologia, Departamento de Obstetrícia e Ginecologia, Faculdade de Medicina FMUSP, Universidade de São Paulo, São Paulo 05508-060, Brazil; 3Escola Superior de Ciências da Santa Casa de Misericórdia de Vitória (EMESCAM), Vitoria 29045-402, Brazil; 4Departamento de Educação Integrada em Saúde, Universidade Federal do Espírito Santo (UFES), Vitória 29040-090, Brazil; 5Department of Urban Planning and Community Development, School for the Environment, University of Massachusetts Boston, Boston, MA 02125, USA

**Keywords:** autonomic nervous system, elderly, exercise, hypertension

## Abstract

Background: High blood pressure is an important public health problem due to its high prevalence, the difficulty to control it, and its high contribution to morbidity. A series of changes may be linked to the aging process, compromising cardiac conduction, and reducing cardiovascular baroreceptor function. Advancing age promotes a decline in heart rate variability and this decrease can increase the probability of cardiovascular disease. The aim of this study was to analyze the autonomic modulation of heart rate in hypertensive elderly individuals during and after a session of aerobic exercise, and to compare it with elderly individuals without cardiovascular or metabolic disease. Our study was a non-randomized controlled study with hypertensive elderly (HBP group) and elderly without cardiovascular and/or metabolic diseases (control group). Data on blood pressure and heart rate variability (HRV) were collected before, during, and after 30 min of aerobic physical exercise on a treadmill. There was a reduction in HF (ms^2^) and SD1 (ms) in the 5 min of recovery for the elderly in the control group. The elderly in the control group also had greater RMSSD and SD1 30 min post-exercise when compared to the initial mins of recovery. We concluded that there was no difference in autonomic modulation and global heart rate variability between elderly individuals without cardiovascular and metabolic diseases and hypertensive individuals after a bout of aerobic exercise. Elderly individuals without metabolic diseases showed a decrease in parasympathetic modulation and global variability between the time of rest and 5 min of recovery. However, up to 30 min of post-exercise recovery, they restored parasympathetic activity.

## 1. Introduction

Cardiovascular diseases are still the biggest cause of death in the world. High blood pressure (HBP), one of the main causes, has been the focus of global health policies to improve populations’ survival. HBP is an important public health problem due to its high prevalence, difficulty in controlling it, and its high cause of morbidity [1]. The elderly population is affected by HBP, and drug treatment associated with a healthy lifestyle are the most successful therapies [1,2,3,4].

With advancing age, the increase in blood pressure has been related to damage to the mechanisms of the autonomic nervous system and the practice of aerobic physical exercise represents an adequate intervention to combat such changes [5,6,7].

Aerobic training (AT) consists of a wide range of exercises practiced around the world by the elderly, and it has been proposed as a tool to improve the baroreceptor reflex and vagal modulation, resulting in an improvement in heart rate variability (HRV) after physical activity [8]. Furthermore, in the elderly, AT has an important antiarrhythmic effect, in addition to other effects on RR intervals, because the practice of AT can change the neuroregulatory control of the heart, improving the influence of the vagal component on hemodynamic parameters [9].

The reactivation of parasympathetic activity can be measured by heart rate recovery (HRR), which is the reduction in heart rate (HR) after physical activity. The HRR can be an important predictor of mortality risk due to its importance in reducing sympathetic activity triggered by exercise [10].

A series of changes can be linked to the aging process, including compromised cardiac conduction and reduced cardiovascular baroreceptor function [11]. However, exercise can improve and combat the physiopathology of cardiovascular aging and the autonomic nervous system of hypertensive and healthy elderly [12]. Thus, the analysis of HRV is another tool for cardiovascular assessment of the autonomic nervous system in the elderly. The influence of aerobic physical activity on autonomic modulation may be one of the fundamental particularities of elderly hypertensive individuals.

HBP has been linked with loss of autonomic modulation of HR, characterized by sympathetic dominance and often by a reduction in the vagal component, which leads to a decrease in HR and blood pressure variability [13,14,15]. Despite post-training sympathetic and parasympathetic activity being good predictors of cardiovascular morbidity and mortality, results on cardiac recovery in the elderly and their HRV indices are still scarce. We hypothesized that elderly hypertensive individuals have decreased HR autonomic modulation during exercise and do not recover the balance of this modulation even after 30 min of exercise.

Our study aimed to analyze the autonomic modulation of heart rate in hypertensive elderly individuals during and after an aerobic exercise session, and to compare it with elderly individuals without cardiovascular or metabolic disease.

## 2. Materials and Methods

### 2.1. Study Design

Our study was a non-randomized controlled study conducted from June 2014 to June 2016 in a rehabilitation clinic for the elderly, located in the state of São Paulo, Brazil, where participants already practiced regular physical activity (practice light or moderate intensity physical activities for about 20 min a day). We adopted the Risk of Bias In Non-randomized Studies of Interventions (ROBINS-I) tool to reduce the likelihood of bias.

Participants were recruited from an elderly rehabilitation clinic. All 100 elderly patients in the clinic were invited to participate. In total, five patients refused to participate in our research. A total of 95 elderly people were included, but five were excluded. Of those who were excluded from the HBP group, two had lung disease, while three were excluded from the control group because they had knee osteoarthrosis that made it impossible for them to walk properly on a treadmill. Although 90 participants were allocated in 2 groups (45 in each group), only 42 individuals in each group completed the study. In the HBP group, three participants reported health problems on the day of the testing procedure and three in the control group did not attend the clinic after recruitment (Figure 1). The two groups recruited unexpectedly had the same size after random recruitment.

Elderly people were invited to be part of the intervention and control groups. All elderly people were assessed as active according to their level of physical activity, measured by the short version of the International Physical Activity Questionnaire (IPAQ).

We split all participants non-randomly into two groups: a group without cardiovascular and/or metabolic diseases called the “Control Group” (CG), and another group diagnosed with high blood pressure (the HBP Group). Outcomes were measured before, during, and after 30 min of aerobic exercise on a treadmill.

Inclusion criteria for the HBP group included: (1) diagnosis of cardiovascular and/or metabolic disease, (2) HBP controlled with medical clearance for physical activity, (3) practice of regular physical activity, (4) age over 65 years old, and (5) signing of the informed consent form. For the control group, the inclusion criteria included: (1) no diagnosis of cardiovascular and/or metabolic diseases, (2) practice of regular physical activity, and (3) age over 65 years old. Exclusion criteria for the CG included: (1) lung diseases, (2) concomitant neurological diseases, (3) the presence of musculoskeletal disorders that did not allow the use of a treadmill, and (4) the use of any medication that could alter autonomic modulation. Exclusion criteria for the HBP group included: (1) pulmonary diseases, (2) concomitant neurological diseases, (3) the presence of musculoskeletal disorders that did not allow the use of a treadmill, and (4) any recent cardiac event or a hypertensive crisis. They took diuretics and renin-angin-aldosterone system inhibitors, while some individuals used a combination of the two drugs. None of the study subjects took antidepressants.

Our study followed the Brazilian regulations on human subject protection contained in Resolutions 466/12 and 251/97 of the National Health Council. It was approved by the Institutional Review Board of the ABC Medical School (CAEE protocol 02830612.1.0000.0082/#108.260). Brazilian Registry of Clinical Trials (REBEC): #RBR-5mgrttm.

**Figure 1 ijerph-20-01292-f001:**
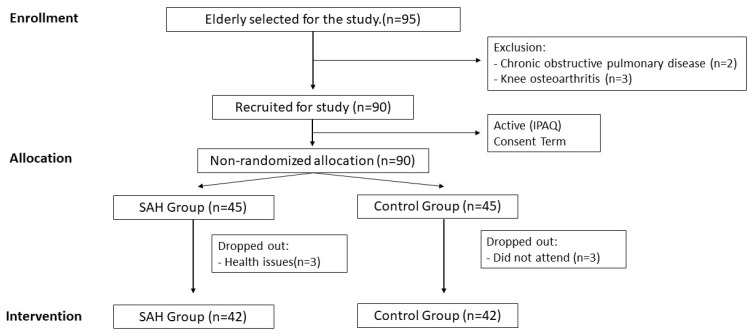
Flow chart of the study design.

### 2.2. Instruments and Data Collection

After signing the Informed Consent Forms, an anamnesis form was used to measure anthropometric variables such as weight and height, and clinical history.

We controlled the ambient temperature (between 21 °C and 23 °C) and humidity (between 40 and 60%), and prepared the equipment before participants arrived at the testing location. Patients were weighed on an electronic scale with a capacity of up to 150 kg and subdivisions of up to 10 g, without shoes and wearing as little clothing as possible. To measure height, patients placed their heels, calves, buttocks, and shoulders against the wall, positioning their heads according to the Frankfurt plane. We used a wall-mounted stadiometer with subdivisions in centimeters and millimeters [16], and calculated the body mass index (BMI) by dividing weight in kg by height squared in meters.

After this first evaluation, the IPAQ was administered. The IPAQ is a scale validated in Brazil and proposed by the World Health Organization in 1998 that is used as a global instrument to determine the population level of physical activity [17,18].

Initially, measurements of systolic (SBP) and diastolic (DBP) blood pressure and heart rate were collected. SBP and DBP data were obtained as a single measurement, indirectly verified by means of a Welch Allyn aneroid sphygmomanometer positioned in the subject’s left arm and a Littmann-3M Classic II stethoscope over the median region of the antecubital fossa. To measure the respiratory rate, a stopwatch was used to count how many thoracic incursions the patient made in one min. HR was recorded using a Polar brand S800CX heart rate monitor. The capture strap was placed in the chest of participants, and the heart rate receiver watch (RS800CX, Polar^®^) (19) was placed on their wrist for HRV analysis.

After placement of the strap and monitor, participants were kept in the supine position and remained at rest for 10 min for baseline HRV capture. Data collection was performed from 1 p.m. to 4 p.m. to avoid changes arising from the circadian cycle in the HRV. After data collection, the heart rate monitor was connected to a wireless receiver computer and the data were transferred to an HRV analysis program (Polar Trainer 5) [19].

We conducted our analysis using the Kubios HRV analysis software (version 2.0.; Kuopio, Finland—University of Eastern Finland—http://kubius.uef.fi, accessed on 12 July 2015) [20,21,22]. HR behavior was recorded beat-to-beat throughout the study. Before using the RR series for data analysis, digital filtering was performed to eliminate premature ectopic beats and artifacts. Only series with more than 95% of sinus beats were included [20,21,22]. Only the initial two mins of the recovery period were considered in our analyses, totaling four analyses windows of 30 s each. This analytical method is adequate to assess the reactivation of post-exercise parasympathetic tone. Next, the HRV indices were calculated before exercise (10 min), during exercise (30 min), and after exercise (30 min) [23].

After digital filtering, the signal was divided into five moments “Rest” originated from a period of 10 min of rest and “exercise” counted as 20 min of the exercise period, that is, the first and last five mins of the 30 min of exercise were disregarded. This cut was made to remove the warm-up and deceleration period of the heart rate, characterizing only the exercise period in the elderly aerobic training zone. “Recovery” consisted of the patient’s recovery period post-exercise. The heart rate variability in this recovery period was divided into three new periods: first (Recovery 1 or R1) within the initial five mins of recovery, second (Recovery 2 or R2) between the tenth to the fifteenth min, and third (R3) from the twenty-fifth to the thirtieth min [16,23]. For the post-exercise period, the measures of SBP, DBP, and RR were also collected in the first, third, and fifth mins, and after the fifth min every 5 min until the thirtieth min.

Patients after a 10-min rest were directed to aerobic training (5-min jogging and 25-min running) on a treadmill for 30 min, maintaining the stipulated training heart rate (70% of the maximum HR). After 30 min, participants were directed to rest in the supine position for another 30 min for the data collection on their recovery time.

### 2.3. Analysis of Heart Rate Variability

#### 2.3.1. Time Domain (TD)

In the time domain, the statistical parameters used to determine the corresponding RR intervals at any point in time [22,24,25] were: (1) the square root of the mean square of the differences between the adjacent normal RR intervals (RMSSD), (2) the standard deviation of all normal RR intervals recorded in a time slot (SDNN) both in ms (ms), and (3) the percentage of adjacent RR intervals with duration difference greater than 50 ms (PNN50). The RMSSD and PNN50 indices predominantly represent parasympathetic modulation. The SDNN represents measures of both branches of the autonomic nervous system and does not allow for distinguishing when changes are due to the removal of vagal tone or increase in sympathetic tone [22,24,25].

The triangular index (RRTri) was calculated from a density histogram of the normal RR intervals. The RRtri index was calculated from the density histogram of the normal RR intervals, obtained by dividing the integral of the histogram by the maximum density distribution, i.e., the modal frequency of RR intervals [22,24,25].

The triangular interpolation of RR intervals (TINN) is the baseline width of the distribution measured as the base of a triangle, approximating the distribution of all RR intervals, in which the least squares difference was used to determine the triangle [22,24,25]. The standard deviation of instantaneous beat-to-beat variability (SD1) represents the dispersion of points perpendicular to the line of identity and appears to be an instantaneous recording index of beat-to-beat variability. The standard deviation of the projection of the Poincaré plot on the line of identity (SD2), i.e., the long-term standard deviation of continuous RR intervals, represents the dispersion of points along the identity line and the HRV in long-term recordings. SD1 and SD2 are expressed in ms. The SD1/SD2 ratio measures the ratio between the short and long variations of the RR intervals [22,24,25].

#### 2.3.2. Frequency Domain (FD)

We used the spectral components low frequency (LF), high frequency (HF) in ms squared, and LF/HF ratio for the analysis of HRV in the frequency domain. To obtain the spectral indices, the frequency tachogram undergoes mathematical processing and creates a graph that expresses the variation of RR intervals as a function of time [22,24,25]. Frequency domain analyses of the HRV signal were conducted using the Fast Fourier transform (FFT). We used the FFT-based Welch’s Periodogram method to transform the time series data into frequency data. The HF index corresponds to respiratory modulation and is an indicator of the action of the vagus nerve in the heart. The LF component is due to the joint action of the vagal and sympathetic components on the heart, while the LF/HF ratio reflects the absolute and relative changes between the sympathetic and parasympathetic components of the autonomous nervous system and characterizes the sympathovagal balance of the heart [22,24,25].

### 2.4. Statistical Analysis

Data are presented as group mean values and standard deviations. A multivariate analysis of variance (MANOVA) was used to analyze differences between study groups in all baseline variables. Excel^®^ and SPSS^®^ (Statistical Package for Social Research) 20.0 (Chicago, IL, USA) software for statistical analysis were used to create the database. Descriptive statistics were conducted to calculate measures of central tendency and dispersion. The Shapiro–Wilk test was used to verify data normality.

ANOVA analysis of dependent variables were used for the HBP and control groups by five moments (rest, exercise, R1, R2, and R3) with repeated measures on the last factor. Post-hoc comparisons were carried out using the Tukey-HSD (Honest Significant Difference) test. Bonferroni corrections were not necessary because our two-by-two study design did not demand multiple testing. The significance level was set at *p*  <  0.05. All values of *p* refer to comparisons with the resting (R) condition.

## 3. Results

There were 84 participants into the study, 42 in the control group and 42 in the HBP group. Baseline characteristics of those groups are described in Table 1. There were no significant differences in sex, age, height, weight, body-mass index, and smoking factors between groups. The main effect for groups while exercising on a treadmill was not found for mean HR, suggesting that the mean HR was more similar for the HBP group than for the control group. All study subjects reached 70% of maximal HR (67.54 ± 0.59 vs. 68.75 ± 10.67 in rest and 109.69 ± 12.21vs 106.5 6 ± 6.28 in exercise, *p* = 0.715 and 0.191, respectively), suggesting that the level of aerobic exercise was comparable between both groups (Table 1). The elderly in the SAH group took diuretics (20%) and renin-angin-aldosterone system inhibitors (45%) while some individuals took a combination of the two drugs (35%). The elderly in the CG did not take any type of medication that could alter autonomic modulation.

MANOVA revealed significant effects for groups [Wilks’ lambda = 0.220, F12, 71 = 20.94, *p* < 0.001, ηp^2^ = 0.78, po = 1], moments [Wilks’ lambda = 0.022, F48, 35 = 33.07, *p* < 0.001, ηp^2^ = 0.98, po = 1], and interaction between groups and moments [Wilks’ lambda = 0.079, F48, 35 = 8.49, *p* < 0.001, ηp^2^ = 0.92, po = 1].

The main effect for groups was found only for mean RR, suggesting that the mean RR was higher for the HBP group than for the control group (m = 845 ± 119 ms vs. m = 787 ± 120 ms; *p* = 0.021). The main effects for moments were present in mean RR and PNN50. In other words, in the comparison between rest and exercise, R1, R2, and R3 showed significant differences. The post-hoc test results are also displayed on Figure 2. There were no interactions for group by moments.

The main effects for moments occurred in all frequency domain indexes, suggesting that the comparison between rest and exercise, R1, R2, and R3 had significant differences. The main effects for the group were not found for LF (ms^2^), HF (ms^2^), and LF/HF ratio, which means that such indexes were not higher or lower for the HBP than for the control group. There were interactions for the group by moments for HF (ms^2^); *p* = 0.002. The post-hoc test showed there were no interactions for the group by moments (Figure 3).

**Figure 2 ijerph-20-01292-f002:**
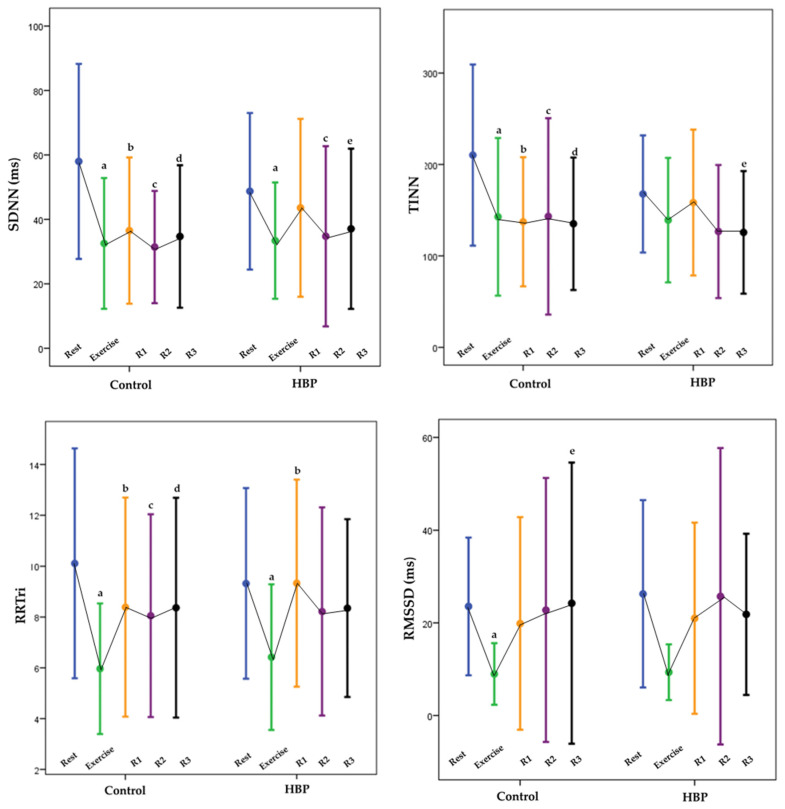
Comparative analysis of HRV (SDNN, TINN, RRTri, and RMSSD) during rest, aerobic exercise, and recovery. MANOVA test (Mean ± SD), HBP (high blood pressure group), Control (Control Group), (R1) recovery period 1—first 5 min of the recovery period, (R2) recovery period 2—between the 10th and 15th min of the recovery period, (R3) recovery period 3—between the 25th and 30th min of the period recovery, (ms) ms, (SDNN) standard deviation of all normal RR intervals recorded at an interval of time, (bpm) beats per min, (RMSSD) root mean square of the square of differences between adjacent normal RR intervals in an interval of time, (TINN) triangular interpolation histogram of RR intervals, (RRtri) triangular index, and (SD) standard deviation. (a) Post hoc test found differences when comparing rest with exercise, (b) post hoc test found differences when comparing rest with R1, (c) post hoc test found differences when comparing rest with R2, (d) post hoc test found differences when comparing rest with R3, and (e) post hoc test found differences when comparing R1 with R3.

The main effects for moments occurred again in all geometrical indexes, except in the TINN (*p* = 0.098), which means that in the comparison between rest and exercise, R1, R2, and R3 had significant differences (Figure 2). There were no significant differences for RRTri, TINN, SD1, SD2, and SD1/SD2 between the HBP and the control group (10.11 ± 4.52 vs. 9.32 ± 3.75, 210 ± 99 ms vs. 1670 ± 64 ms, m = 19.19 ± 19 vs. 18.6 ± 14, 76 ± 41 vs. 65 ± 32 ms, and 0.28 ± 0.25 vs. 0.29 ± 0.18). The post-hoc test results are also displayed on Figure 3. There were no interactions for the group by moments. There were interactions for moments for SD1/SD2 (Figure 3).

**Figure 3 ijerph-20-01292-f003:**
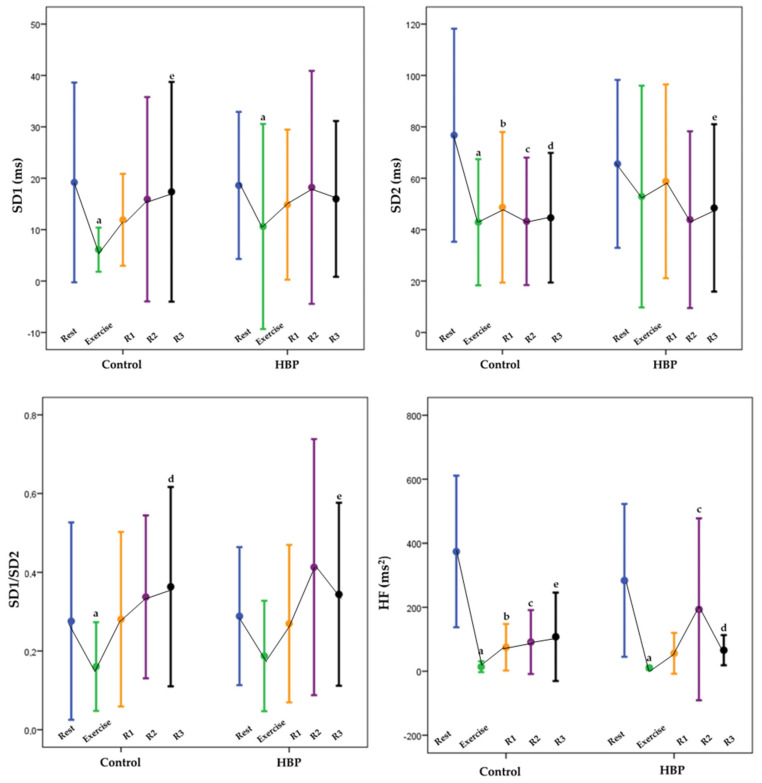
Comparative analysis of HRV (SD1, SD2, SD1/SD2 and HF) during rest, aerobic exercise, and recovery. MANOVA test (Mean ± SD), HBP (high blood pressure group), control (control group), (R1) recovery period 1—first 5 min of the recovery period, (R2) recovery period 2—between the 10th and 15th min of the recovery period, (R3) recovery period 3—between the 25th and 30th min of the period recovery, (SD1) standard deviation of the variability instantaneous beat-to-beat or dispersion of points perpendicular to the line of identity, (SD2) standard deviation of long-term continuous RR intervals or dispersion of points along the line of identity, (SD1/SD2) ratio scatter of points perpendicular to the dispersion line identity of dots along line identity, percentage of adjacent RR intervals with a difference of duration greater than 50ms, (%) percentage, and (HF) high frequency component. (a) Post hoc test found differences when comparing rest with exercise, (b) post hoc test found differences when comparing rest with R1, (c) post hoc test found differences when comparing rest with R2, (d) post hoc test found differences when comparing rest with R3, and (e) post hoc test found differences when comparing R1 with R3.

## 4. Discussion

The interpretations of our results suggest that there was no difference in the autonomic modulation of HR between the elderly participants without cardiovascular and metabolic diseases (the control group), and the hypertensive group. However, there was a reduction in parasympathetic modulation between the pre-exercise moment (rest) and the five mins of recovery in the control group. Although we hypothesized that the autonomic modulation of heart rate in hypertensive participants would be lower than that of participants without cardiovascular or metabolic pathologies, our results show that the elderly in the control group had greater parasympathetic modulation (RMSSD, HF, and SD1) of HR 30 min post-exercise compared to the initial mins of recovery. Furthermore, there was no difference in the global variability of HR between participants without cardiovascular and metabolic diseases and those with hypertension. The aging process causes reductions in the parasympathetic activity of the heart, a fact that promotes a decrease in heart rate variability indices [26,27]. The reduction in cardiac vagal tone, which reflects low parasympathetic activity, is related to autonomic dysfunction, chronic-degenerative diseases, and an increased risk of mortality [28].

In our study, elderly participants with hypertension and those without cardiovascular and metabolic diseases had similar global HR variability indices (SDNN, RRTri, TINN, and SD2), with no differences in parasympathetic modulation indices (RMSSD, HF, PNN50, and SD1). Distinct results were described by Andrade et al. [15] in a study about the impact of arterial hypertension on heart variability in the elderly. The authors reported that there was a decrease in global heart rate variability (lower values of SDNN, TINN, and SD1/SD2 ratio) and a reduction in parasympathetic modulation in the hypertensive elderly, who had lower HF values, with no change in LF.

Perhaps the difference between the two study results may be explained by the intensity of physical activity by the elderly with hypertension in each study because physical training can normalize the sympathetic autonomic modulation and balance the sympathovagal system, which may positively affect the prognosis of cardiovascular diseases [29]. In non-sedentary individuals, physical training promotes an increase in parasympathetic activity and that high aerobic capacity leads to greater parasympathetic activation [29,30]. In the elderly, aerobic training promotes an increase in heart rate variability. Long-term practice of sports with increased total daily energy expenditure and physical activity intensity is associated with higher global HRV and vagal rates. Therefore, it can better counteract the age-related decline in cardiac autonomic control than in sedentary elderly [26].

Masroor et al. (2018) found that in sedentary hypertensive women, four weeks of combined aerobic training with exercises performed at 50–80% of the maximum heart rate, HR training similar to that in our study, was associated with an increase in HF and RMSSD, and a decrease in LF/HF, demonstrating improvement in cardiac autonomic control. Furthermore, combined aerobic and resistance training promoted significant improvement in HRV parameters, indicating vagal dominance in middle-aged hypertensive women [31].

Within the context of physical activity, it is essential to verify not only the heart rate response at the beginning of exercise, which expresses the integrity of the vagus nerve, but also the recovery of HR in the post-exercise moments, because it provides important prognostic information. After all, individuals who have slow HR recovery in the initial mins post-exercise have a high risk of mortality [28].

We found that during the initial 5 min of recovery (R1), the elderly without high blood pressure and metabolic diseases showed a decrease in parasympathetic modulation (HF and SD1) in relation to rest, but those parameters were restored to levels like those at rest only after 30 min post- exercise. In addition. we observed that the HF interacted between groups and moments (*p* = 0.002) at 30 min after exercise. The elderly in the HBP group had reduced levels of this parameter, both in relation to the control group and to rest. One of the hypotheses for this finding is that there may have been reduced venous return and deactivation of cardiopulmonary receptors, since post-exercise hypotension was not observed [32].

The control group had lower parameters associated with global variability in the five mins of recovery (SDNN, RRTri, and TINN), which remained small even after 30 min of recovery, compared to the rest moment. However, there was an increase in SD1/SD2 indices at 30 min post-exercise. On the other hand, the elderly without hypertension and metabolic diseases did not recover from the increase in sympathovagal balance, with LF/HF indexes higher than at rest, even 30 min after exercise, and had a decrease in cardiac autonomic modulation. Differences found during the post-exercise recovery period were described in a study that evaluated the impact of aerobic exercise on heart rate variability in male adults with a mean age of 20 years. The authors reported that heart rate gradually decreased during recovery without reaching pre-exercise values within 30 min post-training, they also found that there was a gradual increase in HRV and that the time and frequency domain indices continuously increased during recovery but remained at reduced values compared to supine rest for at least 30 min [33].

Regarding the sympathovagal balance, elevation of the LF/HF index in individuals who do not have high blood pressure may be associated with increase in blood pressure during the performance of activities, with impairment of the HRV that indirectly reflect on the system autonomic nervous [27,34].

The reduction in parasympathetic and autonomic modulation verified in the 5 min post-exercise in the control but not in the HBP group may be related to the specific characteristics of the elderly in the HBP group. The HBP group in our study consisted of physically active elderly who had controlled blood pressure. Thus, it is possible that the use of medication to control blood pressure associated with physical activity and adherence to non-pharmacological therapies contributed to the differences observed between the groups. Lifestyle changes such as a low-sodium diet [35], smoking cessation [36], and regular physical activity have been used as non-pharmacological strategies for hypertensive individuals [36] and may contribute to cardiovascular health.

Furthermore, differences in heart rate recovery may vary in elderly hypertensive individuals depending on BP control. Amaro-Vicente et al. [37] investigated heart rate recovery in normotensive and hypertensive individuals whose mean age was over 57 years using antihypertensive drugs with controlled and uncontrolled blood pressure. They found that in hypertensive patients, the ideal BP after antihypertensive medication is related to an improvement in sympathetic/vagal balance [38].

Although hypertension causes a delay in heart rate recovery after maximal exercise tests, physical training can normalize HR during the post-exercise period in hypertensive patients. Physical training for a duration of four months can restore the post-exercise decline rate of the recovery heart rate in patients with HBP [36].

Despite the benefits of aerobic exercise in improving cardiac autonomic control, clinical supervision is necessary, even for elderly people who do not have hypertension and cardiovascular diseases. Thus, knowing that there are risks of cardiovascular events during and after exercise, monitoring cardiac autonomic modulation by HR variability in periods after aerobic exercise can be an important tool for the elderly, regardless of the presence of cardiovascular and metabolic diseases, such as systemic arterial hypertension.

Knowledge of risks of cardiovascular events during and after exercise and monitoring cardiac autonomic modulation by HR variability in periods after aerobic exercise can be an important tool for the elderly, regardless of the presence of cardiovascular and metabolic diseases, such as systemic arterial hypertension.

### Limitations

The hypertensive elderly participants evaluated in this study were physically active, clinically able to practice physical activity, and had controlled HBP. Thus, our results do not provide information on the heart variability of hypertensive elderly in advanced stages, whose injuries may be more severe. Furthermore, since the HRV parameters were evaluated up to 30 min after recovery, the actual time when all HRV parameters return to their baseline is unknown. Further research should monitor subjects for 24 h post-exercise. Our results were somewhat unexpected because we expected greater differences between cases and controls. Reasons that may justify the effects of aerobic exercise on autonomic balance in elderly hypertensive patients without complications include: a lack of randomization, the control group belonging to a rehabilitation clinic, and physical activity. We do not have any recovery data extending the 30-min observation period, perhaps after an hour or two there might be significant changes. Finally, one of the strengths of this research is that the control group was made up of healthy elderly people who did not take drugs such as diuretics, renin-angin-aldosterone system inhibitors, or beta-blockers. However, we could not compare the two groups in relation to the use of medication.

## 5. Conclusions

There was no difference in autonomic modulation and global heart rate variability between elderly people without cardiovascular and metabolic diseases and hypertensive elderly people after a bout of aerobic exercise. The elderly in the control group showed a decrease in parasympathetic modulation and global variability between the time of rest and 5 min of recovery. However, at 30 min of post-exercise recovery, the elderly in the control group restored parasympathetic activity.

## Figures and Tables

**Table 1 ijerph-20-01292-t001:** Baseline characteristics of participants according to sex, age, height, weight, body mass index, smoking, blood pressure, and heart rate.

Variables	Total n = 84	Control Group n = 42	HBP Group n = 42	*p*-Value
Sex, n female (%)	74.0 (88)	34.0 (81)	40.0 (95)	0.088 ^a^
Age (years), median	68.3 ± 6.2	68.4 ± 6.4	68.3 ± 6.1	0.911
Height (m)	1.61 ± 0.1	1.61 ± 0.1	1.59 ± 0.1	0.122
Weight (kg)	73.4 ± 11.2	75.3 ± 13.1	71.7 ± 8.4	0.155
Body-mass Index, kg/m^2^	28.4 ± 3.2	28.3 ± 4.1	28.4 ± 3.8	0.605
Current Smoker, n (%)	8.0 (9.5)	4.0 (9.5)	4.0 (9.5)	1.000
Resting Heart Rate, bpm	68.1 ± 8.1	67.5 ± 5.5	68.7 ± 10.6	0.715
Exercise Heart Rate, bpm	107.5 ± 9.0	109.6 ± 12.2	106.5 ± 6.2	0.191
Recovery Period Heart Rate, bpm	72.5 ± 6.0	72.6 ± 2.6	73.5 ± 10.0	0.410
Resting SBP	128.0 ± 7.0	125.0 ± 5.0	130.0 ± 5	0.613
Resting DBP	85.0 ± 5.0	85.0 ± 4.0	85.0 ± 5	0.932

Data are expressed as mean and standard deviation. Analyzed by a Fisher test and *t*-test, (BMI) body mass index, (F) female, (HBP) High blood pressure, (m) meter, (Kg) Kilogram, (kg/m^2^) Kilogram per square meter, (bpm) beat per min, (SBP) systolic blood pressure, and (DBP) diastolic blood pressure. ^a^ Female sex was tested in by linear regression and did not affect the outcome of the study.

## Data Availability

The data of this work can be requested from the corresponding author (email: rodrigo.raimundo@fmabc.br).

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
