# Peer review of "Cardiac Behavior and Heart Rate Variability in Elderly Hypertensive Individuals during Aerobic Exercise: A Non-Randomized Controlled Study"

_ijerph, 2023, doi:10.3390/ijerph20021292_

Round 1

Reviewer 1 Report

Evaristo de Andrade et al. analyzed the Heart Rate Variability (HRV) of elderly (>65 years) subjects before and after 5 and 30 minutes from a physical effort, comparing those with arterial hypertension to those without cardiovascular comorbidities. The article reads well, but the section about results must be improved. Furthermore, I have some “formal suggestions” to make.

Abstract

The abstract is too long and should be reduced, focusing more on the objectives and the findings of the study. Moreover, in the methods the authors should specify the study design (cross-sectional).

Materials & Methods

-          It s not clear which kind of patients have been included in the study. What does “regular physical activity” means? How many times per week did the included patients practice physical activity?

-          The sentence “changes in the central nervous system and/or neurological syndrome” is not clear. Did the authors mean “concomitant neurological diseases”?

-          The signing of informed consent should be inserted among the inclusion criteria for both groups.

-          “clinical and laboratory diagnosis with the disease confirmed by a medical report” should be “diagnosis of the disease”.

-          “Kubios HRV analysis software, version 2.0”. The authors should insert a reference or, alternatively, the place in which the software has been produced.

-          For the frequency domain of HRV, the authors should mention which transform was used to compute the HF and LF bands (Fourier? Other?). Moreover, there is no mention about the unit of measurement employed (ms^2? NU? Hz?) in the methods section. Please specify.

-          The authors should provide a list of drug treatments taken by the study participants, since many drugs (i.e. ACE-inhibitors, Sartans, β-blockers, α-blockers, tricyclic antidepressant) have a known impact on the autonomic nervous system and, consequently, on HRV. Alternatively this lack should be inserted among the limitations of the study.

Results

The results of the study are unclear and the authors should report a graphic description of the same.

Discussion

-          There are some typos that should be corrected.

-          The discussion is too long and should focus more on the results emerged from this study.

Author Response

Re: ijerph-2024102

Title: Cardiac behavior and heart rate variability in elderly hypertensive individuals during aerobic exercise: a non-randomized controlled study

Dear Reviewers

Thank you for the careful review of our manuscript. As suggested, we have revised the manuscript to improve it. We really appreciate the kind suggestions and recommendations of the reviewers, which significantly helped us to improve the manuscript. We hope the current version is accepted for publication in the IJERPH.

With our best wishes.

The authors

Note: The new version of the manuscript also contains changes by the other reviewers.

Dear Reviewer 1

We very much appreciate your constructive reviews to our submission. We revised the manuscript to address the issues you raised. The added or modified words, phrases, and sentences are in red.

English language and style

(x) English language and style are fine/minor spell check required

Answer: the English language was revised

Does the introduction provide sufficient background and include all relevant references?

(x) Yes

Are all the cited references relevant to the research?

(x) Yes

Is the research design appropriate?

(x) Yes

Are the methods adequately described?

(x) Can be improved

Answer: The methods section was revised

Are the results clearly presented?

(x) Must be improved

Answer: The results section was revised

Are the conclusions supported by the results?

(x) Can be improved

Answer: The conclusion section was revised

Comments and Suggestions for Authors

Evaristo de Andrade et al. analyzed the Heart Rate Variability (HRV) of elderly (>65 years) subjects before and after 5 and 30 minutes from a physical effort, comparing those with arterial hypertension to those without cardiovascular comorbidities. The article reads well, but the section about results must be improved. Furthermore, I have some “formal suggestions” to make.

Answer: We appreciate the comments. The Andrade et al manuscript was reviewed by the authors

Abstract

The abstract is too long and should be reduced, focusing more on the objectives and the findings of the study. Moreover, in the methods the authors should specify the study design (cross-sectional).

Answer: The abstract was shortened. As there was an intervention (exercise on a treadmill) we considered it an experimental study. This study is registered as a non-randomized clinical trial (#RBR-5mgrttm) - this information appears in the abstract.

(“The abstract should be a single paragraph and should follow the style of structured abstracts, but without headings”)

Materials & Methods

-          It s not clear which kind of patients have been included in the study. What does “regular physical activity” means? How many times per week did the included patients practice physical activity?

Answer: The information was included in the "methods" section

-          The sentence “changes in the central nervous system and/or neurological syndrome” is not clear. Did the authors mean “concomitant neurological diseases”?

Answer: The sentence was changed.

-          The signing of informed consent should be inserted among the inclusion criteria for both groups.

Answer: We accepted the suggestion. The information was included in the "methods" section

-          “clinical and laboratory diagnosis with the disease confirmed by a medical report” should be “diagnosis of the disease”.

Answer: The sentence was changed.

-          “Kubios HRV analysis software, version 2.0”. The authors should insert a reference or, alternatively, the place in which the software has been produced.

Answer: We conducted our analysis using the Kubios HRV analysis software (version 2.0.; Kuopio, Finland - University of Eastern Finland – http://kubius.uef.fi) [20-22]

-          For the frequency domain of HRV, the authors should mention which transform was used to compute the HF and LF bands (Fourier? Other?). Moreover, there is no mention about the unit of measurement employed (ms^2? NU? Hz?) in the methods section. Please specify.

Answer:  We accepted the suggestion. Frequency domain analyses of the HRV signal was made using Fast Fourier transform (FFT) and the units were revised.

-          The authors should provide a list of drug treatments taken by the study participants, since many drugs (i.e. ACE-inhibitors, Sartans, β-blockers, α-blockers, tricyclic antidepressant) have a known impact on the autonomic nervous system and, consequently, on HRV. Alternatively this lack should be inserted among the limitations of the study.

Answer: The suggested information was included in the "methods" section

Results

The results of the study are unclear and the authors should report a graphic description of the same.

Answer: As requested, the main results were readjusted for graphics (Figures 2 and 3)

Discussion

-          There are some typos that should be corrected.

Answer: We accepted the suggestion revised

-          The discussion is too long and should focus more on the results emerged from this study.

Answer: The discussion was reduced

Reviewer 2 Report

Abstract: Please check that word counts meet the submission guideline. I think your abstract should be summarized.

Line 52: High blood pressure HBP?

Line 54-55:Please add some references.

Line 56-58, 58-60: What do you want to state in this paragraph? These statements indicate different content and seem to have little relevance. I recommend reorganizing the paragraph.

Line 70: HRheart rate (HR)

Introduction: Please reconsider paragraph construction throughout the introduction. It seems to be overly subdivided.

Line 98: All elderly HBP patients over 65 years old and elderly without cardiovascular and/or metabolic diseases were invited to be part of the control group. Although you state about the control group, how was the intervention group?

Line 101: Did you split the control group into 2 group?

Line 105-112:I think the explanation in Line 97 (over 65 years old, without cardiovascular and/or metabolic diseases) is included in this paragraph. In other words, it is partially duplicated. I recommend that it be organized.

Line 116-118: Were five patients excluded after or before allocation. I think the Figure1 and the text do not match.

Figure 1: What is SAH group?

Figure 1: Were the two groups recruited so that there were half in the two groups, or did they unexpectedly to equalize after random recruitment?

Line 114-122 : This paragraph should be moved to the beginning of Methods. In other words, the methods should be structured based on the flow shown in Fig1.

Line 140-142 and 99-100: These statements are duplicative. 

Line 140:What was this scale used for in this study?

Line 144: HR

Line 178: Please describe aerobic exercise on a treadmill in detail. What type of aerobic exercise?

Line 182: Analysis of Heart Rate Variability:I am not an expert in this area, but please double check throughout.

Line 231-232: hypertensive group → HBP group. Please check abbreviations throughout the paper.

Table 1: Does the fact that most of the participants are female cause any impact on the outcome of this study?

Table 1:  Please list the decimal point.

Table 2-4: Please unify the decimal points.

Line 280-284: It is difficult to understand the intent of this paragraph. Please consider a structure that makes it easier to understand the relationship between the preceding and following sentences.

I think the Discussion is very well written overall, but the paragraphs are too subdivided and it is difficult to understand the important information stated in each paragraph. I recommend reorganizing the paragraphs throughout.

Author Response

Re: ijerph-2024102

Title: Cardiac behavior and heart rate variability in elderly hypertensive individuals during aerobic exercise: a non-randomized controlled study

Dear Reviewers

Thank you for the careful review of our manuscript. As suggested, we have revised the manuscript to improve it. We really appreciate the kind suggestions and recommendations of the reviewers, which significantly helped us to improve the manuscript. We hope the current version is accepted for publication in the IJERPH.

With our best wishes.

The authors

Note: The new version of the manuscript also contains changes by the other reviewers.

Dear Reviewer 2

We very much appreciate you for your constructive review of our submission. We revised the article to address the issues raised. The added or modified words, phrases, and sentences are in red.

English language and style

(x) I don't feel qualified to judge about the English language and style

Does the introduction provide sufficient background and include all relevant references?

(x) Can be improved

Answer: The introduction was revised

Are all the cited references relevant to the research?

(x) Can be improved

Answer: The references were revised

Is the research design appropriate?

Are the methods adequately described?

(x) Can be improved

Answer: The methods were revised

Are the results clearly presented?

(x) Can be improved

Answer: The results were revised

Are the conclusions supported by the results?

(x) Can be improved

Answer: The conclusions were revised

Abstract: Please check that word counts meet the submission guideline. I think your abstract should be summarized.

Answer: We accepted the suggestion. The abstract was shortened

(“The abstract should be a single paragraph and should follow the style of structured abstracts, but without headings”)

Line 52: High blood pressure →HBP?

Answer: We accepted the suggestion.

Line 54-55:Please add some references.

Answer: References were added.

Line 56-58, 58-60: What do you want to state in this paragraph? These statements indicate different content and seem to have little relevance. I recommend reorganizing the paragraph.

Answer: We rewrote the paragraph as suggested.  

Line 70: HR→heart rate (HR)

Answer:  We accepted the suggestion.

Introduction: Please reconsider paragraph construction throughout the introduction. It seems to be overly subdivided.

Answer: Thanks for the comment. Some paragraphs were reconstructed.

Line 98: All elderly HBP patients over 65 years old and elderly without cardiovascular and/or metabolic diseases were invited to be part of the control group. Although you state about the control group, how was the intervention group?

Answer: The paragraph was revised

Line 101: Did you split the control group into 2 group?

Answer: No, We didn´t. The paragraph was revised.

Line 105-112:I think the explanation in Line 97 (over 65 years old, without cardiovascular and/or metabolic diseases) is included in this paragraph. In other words, it is partially duplicated. I recommend that it be organized.

Answer: The paragraph was revised.

Line 116-118: Were five patients excluded after or before allocation. I think the Figure 1, and the text do not match.

Figure 1: What is SAH group?

Figure 1: Were the two groups recruited so that there were half in the two groups, or did they unexpectedly to equalize after random recruitment?

Answer: The two groups recruited unexpectedly stayed equalized after random recruitment. This information was included in the manuscript

  Figure 1 was revised constructed

Line 114-122 : This paragraph should be moved to the beginning of Methods. In other words, the methods should be structured based on the flow shown in Fig1.

Answer: The paragraph was moved to the beginning of the methods section.

Line 140-142 and 99-100: These statements are duplicative.

Line 140:What was this scale used for in this study?

Answer: The paragraph was revised

Line 144: HR

Answer:  We accepted the suggestion.

Line 178: Please describe aerobic exercise on a treadmill in detail. What type of aerobic exercise?

Answer: We revised the text to read as follows. Patients after a 10-minute rest were directed to aerobic training (5-minute jogging and 25-minutes running) in a treadmill for 30 minutes, maintaining the stipulated training heart rate (70% of the maximum HR).

Line 182: Analysis of Heart Rate Variability:I am not an expert in this area, but please double check throughout.

Answer: The section was revised.

Line 231-232: hypertensive group → HBP group. Please check abbreviations throughout the paper.

Answer: The change was made.

Table 1: Does the fact that most of the participants are female cause any impact on the outcome of this study?

Table 1:  Please list the decimal point.

Female sex was tested in a linear regression and did not affect the outcome of the study.

Table 2-4: Please unify the decimal points.

Decimals have been unified. For the values of "p" three decimal places were used.

Line 280-284: It is difficult to understand the intent of this paragraph. Please consider a structure that makes it easier to understand the relationship between the preceding and following sentences. I think the Discussion is very well written overall, but the paragraphs are too subdivided and it is difficult to understand the important information stated in each paragraph. I recommend reorganizing the paragraphs throughout.

Answer:  We revised the paragraph

Reviewer 3 Report

The authors performed a non-randomized controlled study on 42 otherwise uncomplicated elderly patients with HT and 42 controls aiming at comparing autonomic HR modulation and HRV before, during and in the recovery phase of a session of aerobic exercise on a treadmill (30 min at 70% max HR). Measurements (Polar brand S800CX, RS800CX, Polar Trainer 5, Kubios HRV analysis) were performed at rest, in the mid-part (20 min) of exercise and 3 times during recovery (R1: <5 min, R2: 10-15 min, R3: 25-30 min). Study variables of HRV comprised both conventional time- and frequency-domain parameters: respectively RMSSD, SDNN, PNN50, RRTri, TINN, Poincaré plot SD1/SD2 and HF and LF.  

In essence there were no significant differences in autonomic HR modulation and HRV between cases and controls except for a few selected conditions: reduction of parasympathetic modulation at rest and at R1 in the controls as well as a greater parasympathetic modulation at R3 versus R1 in the controls.

Comments

1.      We are dealing with interesting results on the effects of aerobic exercise on autonomic balances in elderly otherwise uncomplicated hypertensive patients. The results are somewhat unexpected. The reviewer expected larger differences between cases and controls. Why did not the authors observe the expected differences: lack of randomization, low power, drug therapy, different level of treadmill exercise? The author should critically analyze all possibilities.

2.      Major issue. More specific data are needed on the performance on the treadmill by both cases and controls to demonstrate that the level of aerobic exercise was comparable between both groups.

3.      Do the authors have recovery data extending the 30 min observation period e.g. at one or two hours? Maybe we are dealing with somewhat delayed recovery at 30 min versus controls. Is there evidence that somewhat delayed recovery after aerobic exercise would carry a compromised prognosis in HT?

4.      Didactic issue. The authors may consider a Figure depicting some of the results illustrating the lack of difference in autonomic HR modulation in patients versus controls.

5.      The two sub-analyses yielding significant differences (Abstract) should be taken with caution. Indeed there might be a physiology-based explanation for those differences but physiology-based explanations might hold as well for some of the non-significant associations a priori deemed significant in the reviewer’s mind.

6.      Effects of drug therapy. What is known about the drug therapy in the patients? What is known (or expected) on the effects of drug therapy (Discussion) on autonomic HR modulation?

7.      Have the patients even been advised by their physicians to perform (aerobic) exercise? Maybe there was a training effect induced by whatever kind of non-supervised training in the patients nullifying the expected differences versus (untrained) controls.

Author Response

Re: ijerph-2024102

Title: Cardiac behavior and heart rate variability in elderly hypertensive individuals during aerobic exercise: a non-randomized controlled study

Dear Reviewers

Thank you for the careful review of our manuscript. As suggested, we have revised the manuscript to improve it. We really appreciate the kind suggestions and recommendations of the reviewers, which significantly helped us to improve the manuscript. We hope the current version is accepted for publication in the IJERPH.

With our best wishes.

The authors

Note: The new version of the manuscript also contains changes by the other reviewers.

Dear Reviewer 3

The authors conducted a non-randomized controlled study of 42 otherwise uncomplicated elderly patients with HT and 42 controls aiming at comparing autonomic HR modulation and HRV before, during and in the recovery phase of a session of aerobic exercise on a treadmill (30 min at 70% max HR). Measurements (Polar brand S800CX, RS800CX, Polar Trainer 5, Kubios HRV analysis) were performed at rest, in the mid-part (20 min) of exercise and 3 times during recovery (R1: <5 min, R2: 10-15 min, R3: 25-30 min). Study variables of HRV comprised both conventional time- and frequency-domain parameters: respectively RMSSD, SDNN, PNN50, RRTri, TINN, Poincaré plot SD1/SD2 and HF and LF.  In essence there were no significant differences in autonomic HR modulation and HRV between cases and controls except for a few selected conditions: reduction of parasympathetic modulation at rest and at R1 in the controls as well as a greater parasympathetic modulation at R3 versus R1 in the controls.

We very much appreciate you for your constructive reviews to our submission. We revised the material to address the issues raised. The added or modified words, phrases, and sentences are in red.

English language and style

(x) English language and style are fine/minor spell check required

Answer: The manuscript was revised

Does the introduction provide sufficient background and include all relevant references?

(x) Yes

Are all the cited references relevant to the research?

(x) Yes

Is the research design appropriate?

(x) Can be improved

Are the methods adequately described?

Answer: The methods section was revised

(x) Can be improved

Are the results clearly presented?

(x) Can be improved

Answer: The results section was revised

Are the conclusions supported by the results?

(x) Can be improved

Answer: The conclusions were revised

Comments

  1. We are dealing with interesting results on the effects of aerobic exercise on autonomic balances in elderly otherwise uncomplicated hypertensive patients. The results are somewhat unexpected. The reviewer expected larger differences between cases and controls. Why did not the authors observe the expected differences: lack of randomization, low power, drug therapy, different level of treadmill exercise? The author should critically analyze all possibilities.

Answer: Some of the suggested information was included as study limitations and in the methods section. We agree that the results were somewhat unexpected. We expected greater differences between cases and controls. Some of the reasons that may justify the effects of aerobic exercise on autonomic balance in elderly hypertensive patients without complications include: lack of randomization, the control group belonging to a rehabilitation clinic, and physical activity. The treadmill exercise activity level was regulated at 70% of the heart rate as per the session method. More details on drug therapy were also  added.

  1. Major issue. More specific data are needed on the performance on the treadmill by both cases and controls to demonstrate that the level of aerobic exercise was comparable between both groups.

Answer: HR and blood pressure values were included in table 1 demonstrating that the level of aerobic exercise was comparable between the two groups.

  1. Do the authors have recovery data extending the 30 min observation period e.g. at one or two hours? Maybe we are dealing with somewhat delayed recovery at 30 min versus controls. Is there evidence that somewhat delayed recovery after aerobic exercise would carry a compromised prognosis in HT?

Answer: We have no recovery data extending the 30-minute observation period. This information was placed in the "study limitations" section

  1. Didactic issue. The authors may consider a Figure depicting some of the results illustrating the lack of difference in autonomic HR modulation in patients versus controls.

Answer: As requested, the main results were readjusted for graphics (Figures 1 and 2)

  1. The two sub-analyses yielding significant differences (Abstract) should be taken with caution. Indeed, there might be a physiology-based explanation for those differences but physiology-based explanations might hold as well for some of the non-significant associations a priori deemed significant in the reviewer’s mind.

Answer: The abstract was shortened due to journal rules. Indeed, the two sub analyses that produced significant differences were highlighted in the manuscript. Physiology-based explanations for those differences, and for those that did not, were discussed in the "discussion" section. The section was shortened to address a suggestion of another reviewer  

  1. Effects of drug therapy. What is known about the drug therapy in the patients? What is known (or expected) on the effects of drug therapy (Discussion) on autonomic HR modulation?

Answer: Information on drug use (diuretics drugs and renin angina aldosterone system inhibitor) was entered in the "methods" section. Our patients did not use beta-blockers, justifying the difficulty in finding a sample for comparison. There are paragraphs in red in the discussion and introduction sections referring to effects of drug therapy. For example, references 37 and 38 address this issue.

  1. Have the patients even been advised by their physicians to perform (aerobic) exercise? Maybe there was a training effect induced by whatever kind of non-supervised training in the patients nullifying the expected differences versus (untrained) controls.

Answer: As mentioned in the first paragraph of the methods section, the patients were recruited from a rehabilitation clinic for the elderly. In addition, we used the Risk of Bias In Non-randomized Studies of Interventions (ROBINS-I) tool to reduce the likelihood of bias. The healthy elderly in the control group were assessed by the level of activity practice measured using the short version of the IPAQ questionnaire. This information was reported in the methods section and mentioned as a study limitation.

Round 2

Reviewer 1 Report

The authors greatly improved the manuscript, I have only one minor concern:

-          The authors should clearly specify how many patients of the control group and of the HBP group were taking each kind of medication (i.e. diuretics; renin-angin-aldosterone system inhibitors, others) and whether there were statistically significant differences in the proportion of patients taking these drugs between groups. Moreover, “renin angina aldosterone system inhibitor” should be “renin-angin-aldosterone system inhibitors”.

Author Response

Re: ijerph-2024102

Title: Cardiac behavior and heart rate variability in elderly hypertensive individuals during aerobic exercise: a non-randomized controlled study

Dear Reviewer

Thank you for the careful review of our manuscript. As suggested, we  revised the manuscript to improve it. We appreciate the kind suggestions and recommendations of the reviewers, which significantly helped us to improve the manuscript.

With our best wishes.

The authors

Comments and Suggestions for Authors

The authors greatly improved the manuscript, I have only one minor concern:

Answer: Dear Reviewer 1

We very much appreciate your constructive reviews to our submission. We revised the manuscript to address the issues you raised. The added or modified words, phrases, and sentences are in red.

- The authors should clearly specify how many patients of the control group and of the HBP group were taking each kind of medication (i.e. diuretics; renin-angin-aldosterone system inhibitors, others) and whether there were statistically significant differences in the proportion of patients taking these drugs between groups.

Answer: The elderly in the control group did not take any medication that could alter autonomic modulation. To address the comments of the reviewer clearer, the following information was included in the manuscript: (a) exclusion criteria for the control group (b) percentage of use of diuretics and renin-angin-aldosterone system inhibitors, and (c) a limitation of the study for comparing the two groups in relation to use of medications.

- Moreover, “renin angina aldosterone system inhibitor” should be “renin-angin-aldosterone system inhibitors”.

Answer: The sentence has been changed.

Reviewer 3 Report

The authors have addressed all questions and comments which have been raised by the reviewer and adequately revised their manuscript. The reviewer does not have further comments.

Author Response

Re: ijerph-2024102

Title: Cardiac behavior and heart rate variability in elderly hypertensive individuals during aerobic exercise: a non-randomized controlled study

Comments and Suggestions for Authors

The authors have addressed all questions and comments which have been raised by the reviewer and adequately revised their manuscript. The reviewer does not have further comments.

Dear Reviewer

Thank you for the careful review of our manuscript.

With our best wishes,

The authors